# Antibiotics and the Nervous System—Which Face of Antibiotic Therapy Is Real, Dr. Jekyll (Neurotoxicity) or Mr. Hyde (Neuroprotection)?

**DOI:** 10.3390/molecules26247456

**Published:** 2021-12-09

**Authors:** Magdalena Hurkacz, Lukasz Dobrek, Anna Wiela-Hojeńska

**Affiliations:** 1Department of Clinical Pharmacology, Wroclaw Medical University, 50-556 Wroclaw, Poland; magdalena.hurkacz@umw.edu.pl (M.H.); lukasz.dobrek@umw.edu.pl (L.D.); 2Clinical Pharmacy Service, Jan Mikulicz-Radecki University Clinical Hospital, 50-556 Wroclaw, Poland

**Keywords:** antibiotics, neurotoxicity, adverse drug reaction, neurotransmission

## Abstract

Antibiotics as antibacterial drugs have saved many lives, but have also become a victim of their own success. Their widespread abuse reduces their anti-infective effectiveness and causes the development of bacterial resistance. Moreover, irrational antibiotic therapy contributes to gastrointestinal dysbiosis, that increases the risk of the development of many diseases, including neurological and psychiatric. One of the potential options for restoring homeostasis is the use of oral antibiotics that are poorly absorbed from the gastrointestinal tract (e.g., rifaximin alfa). Thus, antibiotic therapy may exert neurological or psychiatric adverse drug reactions which are often considered to be overlooked and undervalued issues. Drug-induced neurotoxicity is mostly observed after beta-lactams and quinolones. Penicillin may produce a wide range of neurological dysfunctions, including encephalopathy, behavioral changes, myoclonus or seizures. Their pathomechanism results from the disturbances of gamma-aminobutyric acid-GABA transmission (due to the molecular similarities between the structure of the β-lactam ring and GABA molecule) and impairment of the functioning of benzodiazepine receptors (BZD). However, on the other hand, antibiotics have also been studied for their neuroprotective properties in the treatment of neurodegenerative and neuroinflammatory processes (e.g., Alzheimer’s or Parkinson’s diseases). Antibiotics may, therefore, become promising elements of multi-targeted therapy for these entities.

## 1. Introduction. Antibiotics and Antibiotic-Induced Adverse Drug Reactions

Antibiotics are one of the most widely used classes of drug that have revolutionized the treatment of infectious diseases, enabling the causal treatment of these conditions. The discovery of antibiotic agents and their introduction into clinical practice is considered to be one of the greatest medical breakthrough of the 20th century [1]. Mankind has used antibacterial agents of natural origin since the dawn of its history, based on empirical knowledge and centuries-old tradition in various healing systems (e.g., traditional Chinese medicine and others). Traces of tetracyclines, incorporated into the hydroxyapatite mineral portion of bones, have been found in skeletal remains of ancient people (e.g., of the Roman period or even in Sudanese Nubian human remains dated back to 350–550 CE) [2]. The beginning of the modern “antibiotic era” and antibiotic therapy used to treat human infections is usually associated with names of Paul Ehrlich, Gerhard Domagk and Alexander Fleming. These researchers became famous in the history of medicine with the introduction of the first modern, arsenic-based antimicrobial agent named Salvarsan, effective in the treatment of syphilis (Ehrlich; 1909), the discovery of the sulfa drug, sulfonamidochrysoidine (Protonsil), endogenously releasing active sulfanilamide (Domagk; 1935; Nobel Prize laureate in Medicine or Physiology in 1939 for the development of antibacterial effect of Protonsil) and the discovery of penicillin (Fleming; 1929; Nobel Prize laureate in Medicine or Physiology in 1945 for the discovery of penicillin and its curative effect in various infectious diseases) [2,3]. Obviously, these “milestones” of antibiotic therapy would not have been possible without the prior work of other researchers, such as Antonie van Leeuwenhoek, Robert Hooke, Robert Koch and Louis Pasteur who laid the basics for modern microbiology [3]. Then, “the golden age of antibiotic discovery” began, which lasted for about 20 years and resulted in the introduction of most of the currently used antibiotics in clinical practice. The twilight of this period, which also includes the present times, is the aftermath and one of the fundamental problems of antibiotic therapy, i.e., the development of bacterial strains resistant to various antibiotics, which results in the loss of the anti-infective effectiveness of many of the preparations used so far. Uncontrolled infectious diseases are again becoming an emerging problem in modern medicine. Estimates indicate that mortality rates due to multidrug-resistant bacterial infections have become increasingly higher—each year, about 25,000 of patients treated in the EU die from multidrug-resistant bacterial induced infections and in the USA about 63,000 deaths are caused by hospital-acquired infections [2]. Currently, bacterial resistance is not limited to primary inpatients, but is especially true for outpatients. The ongoing antibiotic resistance crisis is determined by various factors, the most important of which include the excessive and unreasonable antibiotic consumption (due to the fact that in many countries antibiotics use is unregulated and available over the counter without a rational medical recommendation), inappropriate prescribing and extensive agricultural use (primarily to promote growth of housed animals and to prevent infections) [4]. Complex genetic mechanisms, including plasmids, bacteriophages, naked DNA or transposons, are the background for the development of bacterial resistance to antibiotics. Spontaneous mutations are also an important cause, allowing the acquisition of bacterial resistance to antibiotics without the exchange of genetic material between various strains [5]. The problem of widespread bacterial resistance to many, if not most, antibiotics still used in the current therapy is a major threat and disadvantage. New solutions are expected (including the discovery of new structures with antimicrobial activity) that would allow us to be “one step” ahead of bacteria in the fight to control infection. To sum up, despite the undisputed benefits of antibiotics, which have saved millions of lives from the consequences of infectious diseases, the current treatment of these diseases is becoming more and more challenging.

Additionally, a significant problem of antibiotic therapy is the occurrence of many possible antibiotic-related adverse drug reactions (ADR). An adverse drug reaction is regarded as an expected, unwanted, harmful or unpleasant effect attributed to the use of a medication that occurs during its usual clinical use. According to the commonly accepted definition given by Edward and Aronson, “an adverse drug reaction is an appreciably harmful or unpleasant reaction, resulting from an intervention related to the use of the medicinal product, which predicts hazard from future administration and warrants prevention or specific treatment, or alternation of the dosage regimen, or withdrawal of the product”. ADRs appear in both outpatients and hospitalized patients and manifest a wide spectrum of clinical entities, ranging from mild symptoms to life-threatening disorders. Estimates indicate that ADRs occur in 5–10% of cases of hospital admissions and, in as many as 0.1–0.3% ADRs, may be serious and cause death [6,7,8].

Antibiotics, along with antiplatelets, anticoagulants, cytotoxics, immunosuppressants, diuretics or antidiabetics, have been particularly implicated in ADR inducement. Perhaps the most characteristic for antibiotic-related ADRs are the symptoms caused by hypersensitivity and allergic reactions, which most often take the form of skin reactions (rash, hives, itching), but these can also be severe disorders such as angioedema or anaphylactic shock. Other characteristic antibiotic-induced ADRs are complex symptoms originating from the gastrointestinal tract (e.g., nausea, vomiting, bloating, diarrhea/constipation) determined by the altered secretion, absorption and motility due to dysbiosis. There are also reported class-specific (or even compound-specific) antibiotic-related ADRs, for example: the possibility of developing pseudomembranous colitis induced by *Clostridium difficile* colonization after application of antibiotics with broad antibacterial activity, aminoglycoside-associated renal toxicity, fluoroquinolone-related tendonitis and Achilles tendon rupture, myelosuppression after linezolid, cardiac arrythmias induced by macrolides or diffuse interstitial pneumonitis, and pulmonary fibrosis that might be the consequence of nitrofurantoin administration [9,10]. However, many antibiotics are considered to exert hepato- and nephrotoxicity, peripheral blood disorders (anemia, leukopenia, thrombocytopenia) or electrolyte abnormalities. Among other potential ADRs induced during antibiotic therapy, uncommon, but possible, neurotoxicity should also be mentioned, mostly associated with the use of beta-lactams or quinolones. Post-antibiotic neurological disorders are manifested by hearing loss or labyrinthine dysfunction (characteristic for erythromycin and azithromycin), and by ototoxicity and vestibular dysfunction (pathognomonic for aminoglycosides) or by other forms of neurotoxicity, affecting either peripheral or the central nervous system [11]. Generally, drug-induced neurological disorders (DIND) are manifested by a very broad spectrum of disorders, e.g., cerebrovascular disease, delirium, headache, nerve and muscle disorders, movement disturbances, seizure attacks, sleep abnormalities and others [12,13]. They are listed in Table 1 below. Among the potential drugs responsible for the development of DIND, antibiotics should also be mentioned. Most commonly, the above-mentioned aminoglycosides and macrolides are characterized by harmful potential toward the nervous system, but this also applies to quinolones, sulfonamides, penicillin, carbapenems, tetracyclines, oxazolidinones, polymyxins’ and metronidazole. These are listed in the next chapter. The toxic effects of antibiotics on the central nervous system are not as well understood as their other side effects and may be confused with symptoms of various neurological or psychiatric diseases.

The pathogenesis underlying DIND is complex. To damage nervous system structures, a drug or its metabolites must either cross the blood–brain barrier (BBB) or become incorporated into the neuron by peripheral axonal uptake and retrograde axonal transport. The first mechanism is mainly used by lipophilic drugs, and their potential neurotoxicity is obviously exacerbated by already existing damage to the BBB. The direct mechanisms by which drugs may produce neurotoxic effects include impairment of neuronal energy production with subsequent disturbances of ion channels’ functioning, disturbances in synthesis and release of neurotransmitters from neuronal terminals, or noxious effects of cellular structures of neurons exerted by drug metabolites. Neuronal ATP synthesis may be affected not only by hypoxia, ischemia or hypoglycemia but also by drugs that disrupt the metabolism by hindering energy production or enhancing energy consumption, contributing to uncoupling of electron flow and oxidative phosphorylation, oxidative stress or inhibition of adenosine enzymatic breakdown. These disturbances lead to the subsequent intracellular ion entry (Ca^2+^, Na^+^) and release of excitatory glutamate which in the “vicious circle” mechanism intensify already existing damage and activity of Ca^2+^-dependent cellular phenomena. Finally, the release of neurotransmitters (serotonin, noradrenaline, dopamine, acetylcholine) is disturbed and the calcium-dependent apoptotic processes of nerve cells occur. The triggering factors facilitating the DIND involve drug related factors (e.g., polydrug abuse, formation of neurotoxic metabolites during endogenous drug metabolism) and individual related factors (age, gender, gestational drug exposure, antioxidant status, diet) or influence of environmental conditions (chronic stress, temperature, exposure to environmental toxins and pollutions) [14,15].

## 2. Antibiotic-Related Neurotoxicity—A General Outline and Pathogenesis

Antibiotics may be causative agents of peripheral or central nervous system dysfunction. The neurogenic ADRs of antibiotics are more common in elderly patients with kidney and/or liver insufficiency and in patients with preexisting neurological abnormalities. As with other ADRs, antibiotic-induced neurological disorders are potentially reversible as long as they are quickly recognized and corrected.

The risk of post-antibiotic peripheral neuropathy occurs with prolonged administration of some antibiotics, e.g., metronidazole. Seizures, twitching and hallucinations are possible neurological ADRs caused mostly by penicillin, imipenem-cilastin, cephalosporins or ciprofloxacin [11]. However, central nervous system toxicities were also demonstrated for sulfonamides, tetracyclines, chloramphenicol, colistin, aminoglycosides, metronidazole, isoniazid, rifampin, ethionamide, cyclo-serine, and dapsone. Cranial nerve toxicities, manifested by myopia, optic neuritis, deafness, vertigo, and tinnitus, were associated with the use of erythromycin, sulfonamides, tetracyclines, chloramphenicol, colistin, aminoglycosides, vancomycin, isoniazid, and ethambutol. Symptoms of paresthesias, motor weakness or sensory impairment, which are considered to be a clinical manifestation of peripheral neuropathy, were associated with the use of penicillin, sulfonamides, chloramphenicol, colistin, metronidazole, isoniazid, ethionamide, and dapsone. Neuromuscular blockade and weakening of the neuromuscular strength were related to the use of tetracyclines, polymyxins, lincomycin, clindamycin, and aminoglycosides. Antibiotic-related neurotoxicity depends on the dosing schedule and the functional status of the liver and kidneys. There are reports that penicillin G intravenous administration may lead to harmful effect in the central nervous system when given more than 50 million units per day in adults [16]. The maximum recommended dose of imipenem-cilastin in adults with preserved renal function that does not cause neurological disorders is 4 g per day and estimates indicate that seizures occurring in patients using this antibiotic occur in 2% of cases [17]. Similarly, fluoroquinolone use was found to be associated with seizures and headaches in 1–2% of recipients. The other, unusual effects observed in patients treated with fluoroquinolones (ofloxacin, sparfloxacin) included orofacial dyskinesia and a Tourette-like syndrome [18]. Neuromuscular blockade and the possibility of intensification of the action of intraoperative muscle relaxants is the most commonly known neurological ADR of aminoglycosides, but the symptoms were also demonstrated for tetracyclines, polymyxins, lincomycin, clindamycin, although to a much lesser extent. Thus, aminoglycosides should be avoided in patients with inherited neuromuscular disturbances, e.g., myasthenia gravis [16]. Ototoxicity or vestibular dysfunction are also well-known neurological ADRs of aminoglycosides. These disturbances are usually dose- and frequency-dependent and correlated with other risk factors for cranial nerve VIII damage, such as advanced age, fever, anemia, baseline creatinine level and concomitant use of other ototoxic agents (e.g., furosemide, salicylate) [19,20,21]. Macrolides-erythromycin and azithromycin administration may cause the bilateral hearing loss or labyrinthine dysfunction and vertigo, and patients with hepatic insufficiency are especially predestined to develop these disturbances. In most cases, these complications were described as dose-dependent and usually reversible within 2 weeks after discontinuation of the treatment, although there have also been reports of irreversible hearing loss [22,23,24,25].

A summary of the neurotoxic effects related to the use of different classes of antibiotics is given in Table 2 below [26,27].

## 3. Antibiotics and the Dysbiosis of the Gastrointestinal Tract and Its Relation to Neurotoxicity

The human digestive tract is inhabited by many microorganisms that form a specific ecosystem, which includes, among others, bacteria, fungi, yeasts and viruses. All these living microorganisms are collectively named “the microbiota” (this term has replaced the previously used term “microflora”), while the collection of genes of the microorganisms constituting the microbiota is called “the microbiome” (containing about 3 million genes). More broadly, the microbiota is also considered to be a collection of all microorganisms found in the various compartments of the human body, including the above-mentioned gut microbiota and organisms inhabiting the skin, distal urogenital or respiratory systems [28,29,30]. The term “microbiota” was introduced by the Nobel Laureate Joshua Lederberg in 2001 to define the whole system of commensal, syn-biotic and pathogenic microorganisms that share a living space with the human host [31,32].

The microorganisms inhabiting the gut are an integral part of its host’s well-being and the composition of the microbiota is individual and unique for every human being. It is formed during childbirth, but modified after birth by many factors: age, genetic conditions of the host, diet, infections and use of antibacterial drugs or probiotics/prebiotics/symbiotics. Under physiological conditions, the intestine is colonized by approximately 10^13^–10^14^ bacteria represented primarily by *Firmicutes, Bacteroidetes, Actinobacteria, Proteobacteria, Fusobacteria* and *Verrucomicrobia*. In health, all intestinal microorganisms are in a state of dynamic equilibrium known as eubiosis. This state has protective, trophic and metabolic functions in the gastrointestinal tract, but also plays a role in controlling the brain activity and behavior. The phenomenon is widely known as the gut–brain axis (GBA) [33]. This axis is functionally based on communication between the central and the enteric nervous system (mostly via vagus nerve fibers), linking centers of the brain with peripheral intestinal functions. GBA appears to be bidirectional, namely through signaling from gut-microbiota to the brain (“bottom-up”) and from the brain to gut-microbiota by means of neural, endocrine, immune, and humoral pathways. The background for signaling from the gut to the brain are several microbiologically derived metabolites (short-chain fatty acids (SCFAs) butyrate, propionate, and acetate, secondary bile acids, tryptophan metabolites). These agents act primarily through interactions with enteroendocrine cells, enterochromaffin cells and the musical immune system, also resulting in an increased release of cytokines (Il-1β, Il-6, TNF-α). There are two barriers to GBA signaling: the intestinal barrier and the blood–brain barrier (BBB). In conditions of dysbiosis (pathological change of eubiosis) there is an increase in the permeability of the intestinal wall (“leaky gut”) and the transfer of bacterial mediators to the blood and inflammation takes place. Dysbiosis is also associated with disruptive BBB changes since abnormal gut microbiota may affect proper traffic between the circulatory system and the cerebrospinal fluid of the central nervous system. The descending modulation of GBA activity takes place indirectly through changes in the activity of the autonomic nervous system and directly through luminal release of neurotransmitters (catecholamines, serotonin, dynorphins). Autonomic fibers (both sympathetic and parasympathetic, the most important being the vagus nerve) control gut functions including motility, secretion, epithelial fluid maintenance, intestinal permeability and mucosal immune response and these phenomena affect the microbial habitat, thus influencing the composition and activity of microbiota [34,35,36,37].

Abnormalities in gut microbiome and GBA interactions have been implicated in the pathogenesis of functional gastrointestinal disorders (e.g., irritable bowel syndrome; IBS) but also in neurologic and psychiatric entities [33,34]. There is also evidence that the dysbiosis-induced stimulation of the vagal nerve, that conveys information between the gut and the central nervous system, intensifies the expression of neurotrophic elements determining the development of new neurons and synaptic connections, which has been shown to be associated with mood disorders [34,38]. Moreover, it has been demonstrated that bacterial strains residing in the intestines may be implicated in affecting the brain by impacting the production or response to neurotransmitters (GABA, serotonin) [34,39,40]. Abnormal gut microbiota may produce a large number of amyloids and other toxins and act as a source of systemic inflammation, thus contributing to increased risk of Alzheimer’s disease development [34,41]. The intestinal dysbiosis also predisposes to Parkinson’s disease development due to the fact that gut inflammatory pathomechanisms play a significant role in alpha-synuclein misfolding [34,42]. It was also found that higher concentrations of *Enterobacteriaceae* were directly proportional to gait and postural instability [34,43]. Studies also revealed the role of the gut microbiome in amyotrophic lateral sclerosis (ALS). In an experimental ALS mouse model, a tight junction structure, greater intestinal permeability, and an abnormal microbiota profile with lower butyrate-producing bacteria were observed compared to controls. Butyrate, a bacterial metabolic by-product, has been proposed to normalize the gut microbiota, as well as to enhance the lifespan of ALS [34,44]. Abnormalities of the gut microbiota and the “leaky gut” syndrome development that allows bacterial metabolites to cross the intestinal barrier is also considered to be implicated in the pathogenesis of some psychiatric entities. The results of some studies demonstrated that persistent, low-grade inflammation as a result of a “leaky gut” predisposes the host to the development of anorexia nervosa, depression and anxiety conditions [34,45,46].

To sum up, the gut microbiota significantly influences the interaction between the gut and the brain through complex neuroendocrine and immune processes. It has long been known that dysbiosis of the intestinal microbiota is associated with various disorders of the nervous system. It is worth mentioning that the use of antibiotics leads to the rebuilding of microbiota composition and activity. Therefore, the administration of antibiotics, in the context of their effect on GBA, can be viewed dichotomously. On the one hand, by increasing the risk of dysbiosis development when used irrationally, these antibacterial agents may promote neurotoxicity. On the other hand, if properly applied, by eliminating pathogenic bacterial strains, antibiotics can significantly reduce the risk of neurological disorders resulting from GBA abnormalities [47].

## 4. Short Description of the Detailed Neurotoxicity of Particular Classes of Antibiotics

### 4.1. Metronidazole

Metronidazole has been used for decades as a broad-spectrum antimicrobial agent effective in the treatment of anaerobic bacterial and protozoal infections and in Helicobacter pylori eradication [48]. Metronidazole-induced encephalopathy (MIE) was first described in 1977 [49]. The most common reported neurologic disturbances are mild and involve dizziness, headache, confusion, vertigo and insomnia [48]. However, an inappropriate, excessive use of the drug may lead to neurological complications, the severity of which reflects total drug exposure. Therefore, it is recommended to reduce rationally both the duration of treatment and adopted doses while maintaining its effectiveness [50,51]. Neurological complications become more common when the drug is used in a dose exceeding 2 g/day for prolonged time [52]. Severe neuropsychiatric disturbances have been observed in patients treated with metronidazole in a total dose of 42 g for 4 weeks of continuous therapy. However, the symptoms are observed to resolve after the discontinuation of the therapy in most patients [53]. The encephalopathy symptoms may be still present within 1–12 weeks following high-dose metronidazole treatment and the imaging abnormalities resolve between 3 to up to 16 weeks after stopping metronidazole administration [52]. In rare cases, persistent MRI abnormalities of the brain and clinical symptoms of encephalopathy have been reported despite discontinuation of the treatment [54]. Metronidazole use is associated with the risk of both central and peripheral neurotoxicity [48,50,55]. Usually, the metronidazole-induced neurotoxicity is characterized by a gradual onset and mostly affects patients with concomitant renal and/or liver dysfunctions [56,57]. The peripheral neuropathy induced by higher doses of metronidazole manifests by sensi-motor neuropathy and in some patients it may be accompanied by autonomic neuropathy in the form of vasomotor and temperature dysregulations. The reason for sequential involvement of peripheral sensory, motor and finally autonomic fibers evoked by the drug is unknown [52,58]. Moreover, the induction of cerebellar and vestibular system damage with subsequent ataxia after metronidazole, often demonstrated in experimental studies, remains unclear [52]. The proposed pathomechanism of metronidazole-induced neurotoxicity is related to the basic antimicrobial action of metronidazole and is associated with the drug’s ability to generate free radicals. These intermediate molecules may cause oxidation of catecholamines and other neurotransmitters and contribute to production of semiquinone and nitro-anion secondary neurotoxic radicals [52,58]. Moreover, inhibition of neuronal protein synthesis and radical injury to nerve tissue may result in peripheral nerve injury (“axonal swelling” and localized neuronal ischemia with perineural edema) leading to mixed neuropathy symptoms development. The proposed, complementary mechanism of metronidazole neurotoxicity is also the inhibition of GABAergic neurotransmission and modulation of the GABA receptor [50,52,55,58]. The hypothesis is in line with the experimental observations that central metronidazole-associated neurotoxicity is ameliorated by diazepam [59].

### 4.2. Sulfonamides/Trimethoprim

Sulfonamides as antagonists of para-aminobenzoic acid (PABA; folate precursor), were the first antimicrobial agents used in the treatment of infections evoked by many Gram-positive cocci and Gram-negative bacilli. Currently, sulfonamides have diminished importance due to resistance, but are still used in the treatment of certain infections (dysentery, plague, tetanus, typhoid and paradour), and as second-line drugs in the treatment of infections of the urinary tract and the respiratory system. They are also externally used on the skin for the prevention of post-burn infections and topically to treat bacterial conjunctivitis. Moreover, these drugs are still important in the prophylaxis and treatment of some opportunistic infections associated with immunodeficiency in the course of AIDS. At present the most clinically relevant is sulfomethoxazole (SMX) administered with trimethoprim (TMP; a dihydrofolate reductase inhibitor) [60]. SMX/TMP administration produces various ADRs, including gastrointestinal, dermatological disturbances, hematological and hypersensitivity reactions. Some neurological disruptions in the form of tremor and transient psychosis with agitation and visual and auditory hallucinations were also reported [26,27,60]. The first reports of psychiatric disorders related to the use of sulfa drugs appeared as early as 1942 [61]. More recently, Walker et al. [62] confirmed the development of temporary psychosis in 20% of immunosuppressed HIV-negative, renal transplantation patients, with *Pneumocystis jirovecii* infection, treated with SMX/TMP. The symptoms appeared between 3–10 days after initiation of the SMX/TMP administration and resolved within 24 h after discontinuation of the treatment. In another study Lee et al. [63] demonstrated that almost 12% of HIV-infected patients with *P. jirovecii*-induced pneumonia treated intravenously with SMX/TMP presented acute psychosis symptoms after an average of 5 days of the drug administration. The symptoms resolved after discontinuation of treatment or, in some cases, after SMX/TMP dose reduction or change of the route of administration from i.v. to oral while maintaining the applied dose. Moreover, SMX/TMP administration was also reported to be, rarely, associated with the aseptic meningitis development or transient-occurring tremor in immunocompromised patients [64,65].

The mechanisms contributing to sulfonamide-induced acute behavioral changes remain unknown. However, it is postulated that the psychiatric disturbances may be related to the SMX/TMP-dependent deficiency of the tetrahydrobiopterin synthesis. This factor is also utilized in the formations of essential central neurotransmitters, e.g., serotonin or dopamine, thus the resulting disturbances in central neurotransmission may be co-responsible for the generation of symptoms of transient psychosis [66]. Some reports also indicate a relationship between a powerful antioxidant level-glutathione and SMX/TMP neurotoxicity. It is also likely that the decrease of preventive glutathione enables the formation of unstable, neurotoxic sulfonamide by-products. This hypothesis would be in line with the observations that sulfonamide neurological ADRs are often noticed in HIV-infected or geriatric patients with depleted endogenous reserves of glutathione [67,68]. Of note, the SMX/TMP neurotoxicity is less commonly reported in children, probably due to the lower doses used in the therapy, the lack of significant concurrent diseases and drug interactions resulting from polypharmacotherapy [69].

### 4.3. Beta-Lactams

The beta-lactam antibiotics include penicillin, cephalosporins, carbapenems and monobactams. Except for monobactams, this group of antibiotics, together with quinolones, have been reported to account mostly for neurological ADRs development. The risk factors of beta-lactam induced neurotoxicity include: renal dysfunction (both acute and chronic) that decreases drug clearance, a blood level decrease in albumin binding of the antibiotics (e.g., hypoalbuminemia), liver insufficiency with downregulation of the hepatic metabolism by cytochromes P450 system, advanced age, high dosing, previous, concomitant diseases of the nervous system, low birth weight in newborns and all pathological conditions predisposing to increased permeability of BBB [26,27,49,60,70]. The most common potential neurological disturbances attributed to penicillin were abnormalities found in electroencephalograms (with epileptiform discharges), myoclonia, seizures and the presence of disorientation, confusion, delusions or hallucinations [26,27,60,70,71,72]. The convulsant effect of penicillin was first observed in 1945 by Walker and Johnson [27,49]. These antibiotics also stimulate T-cells and are responsible for the occurrence of drug-induced aseptic meningitis (DIAM) [49,72]. Hoigne’s syndrome is a specific neuropsychiatric entity associated with the intramuscular use of procaine penicillin. The incidence of the condition is estimated at about 0.8–16.8/1000 injections [71]. The symptoms include panic attacks, depersonalization, auditory, visual, gustatory and somatosensory hallucinations, which are accompanied by adrenergic overstimulation (tachycardia, high blood pressure, shortness of breath) and possible generalized seizures. The attack usually lasts a few minutes and is preceded by residual asthenia and anxiety [73,74]. The potential underlying mechanism of the phenomena is that of embolic events in brain vessels, secondary to accidental penetration of procaine penicillin in the vascular system during injection or the direct toxic effect exerted by procaine with presumed limbic excitation [75]. Penicillin neurotoxicity is manifested primarily upon intravenous or intrathecal administration [27]. Among the penicillin agents, piperacillin and tazobactam appear to be most potent to produce neurotoxicity symptoms [60,76,77], although ampicillin or benzylpenicillin-induced epileptogenic potential has also been reported in the literature [78,79]. It has been shown that symptoms of encephalopathy can occur 1.5 to 7 days after piperacillin or piperacillin/tazobactam administration [72].

Cephalosporin-induced neurological ADRs are similar to those observed after penicillin administration and include, abnormal electroencephalogram, non-convulsive status epilepticus, myoclonus, chorea-athetosis, seizures and psychotic symptoms [26,27,60,70,71]. The variety of clinical presentation, ranging from simple EEG abnormalities to mental status changes, myoclonus, seizures or even coma have been reported within all four generations of cephalosporins, with the most frequent findings related to cefepime, cefoperazone, ceftazidine, cefuroxime and cefazolin [60,80]. The time to develop encephalopathy ranges from 1 to 10 days after medication initiation, and resolves in 2 to 7 days following discontinuation. Renal failure may be responsible for drug accumulation, which promotes the occurrence of neurotoxic effects. These manifest themselves at serum trough concentrations ranging from 15 to 20 mg/L. Other risk factors for encephalopathy are preexisting brain injury, increased serum concentration and overdose of the drug. Analogous to penicillin, they may mediate DIAM through specific drug-IgG binding in cerebrospinal fluid [49]. Cefazolin, ceftazidime, and cefepime-cephalosporins, with higher GABA_A_ receptor affinity and increased BBB penetration, are thought to be more predisposed to cause neurotoxic symptoms [72]. It should be noted that sepsis and systematic inflammation compromise the integrity of BBB and may make it easier for drugs to overcome it [49]. The probability of neurotoxicity of cefotaxime or ceftriaxone is lower than in the above-mentioned group [27], although the study of Lacroix et al. reported the incidence of serious CNS complications associated with ceftriaxone therapy to be seven times higher than that published in the literature. Reported CNS ADRs between 1995 and 2017 identified ceftriaxone as both the leading cause of hospitalization and life-threatening situations, or even death [81].

Carbapenems including imipenem, meropenem, panipenem, ertapenem, and doripenem are also antibiotics that were demonstrated to share common symptoms of neurotoxicity with other beta lactams. Treatment with carbapenems may induce headache, seizures and encephalopathy [26,27,60,70,71]. Seizure incidence of imipenem was estimated in up to 1.5–2% and this decreases with the newer carbapenems, with a value found for doripenem of 1.1% [82,83]. The pro-convulsive effects may be related to its action on the a-amino-3-hydroxy-5-methyl-isoxazolepropionate (AMPA) and NMDA receptor complexes [49].

There is no unambiguously convincing evidence supporting the significant neurotoxicity of monobactams. The leaflet dedicated to these preparations mentions the possibility of seizures, confusion, dizziness, vertigo, paresthesia or insomnia, but they are reported very rarely [70,71].

The mechanisms responsible for beta lactam neurotoxicity are related to the ability of these drugs to exert inhibitory effects on GABA neurotransmission. This effect is thought to be due to the structural resemblance of the beta lactam ring and its affinity to GABA receptor binding since the degradation of the beta-lactam structure prevents the occurrence of seizures [49,84]. Thus, GABA complex receptor inhibition via competitive (for cephalosporins) or non-competitive affecting of the GABA_A_ subunits is the basic hypothesis for beta lactam neurotoxicity [26,84,85]. The complementary hypotheses raise issues of the release of various cytokines with potential for neurotoxicity and an ability to increase the excitatory action of N-methyl-D-aspartate (NMDA) and alpha-amino-3-hydroxy-5-methylisoxazolepropionate receptors with overactivity of the glutamatergic system and accumulation of neurotoxic metabolites [26,49,80,84,85,86,87,88].

### 4.4. Glycopeptides

It would be difficult to imagine the practice of infectious diseases treatment over the past 20 years without glycopeptide antibiotics. Their safety profile is favorable, although vancomycin (with intravenous use) and teicoplanin can induce sensorineural hearing loss, with possible association to tinnitus, dizziness and vertigo [89,90,91,92]. Penetration of vancomycin into cerebrospinal fluid is poor, but has increased in patients with meningitis. Encephalopathy and mononeuritis multiplex are rarely observed during the use of this drug [93]. The mechanism of vancomycin ototoxicity involves direct damage by the drug to the auditory branch of the eighth cranial nerve [91]. Moreover, an explanation for this toxicity may be oxidative stress, which leads to loss of sensory cochlear cells [94]. Transient or permanent hearing loss has been reported during vancomycin use, especially in patients treated with high doses, those receiving concomitant other drugs with ototoxic effects (e.g., aminoglycosides), with renal dysfunction or those with pre-existing hearing impairment. The risk of hearing loss is greater in elderly patients [91,92]. Ototoxicity with teicoplanin has been observed, but it does not occur often [92]. Currently available data suggests that the second generation lipoglycopeptides, dalbavancin and oritavancin, have no effect on hearing loss or dysfunction [95,96].

### 4.5. Macrolides

Macrolides show a similar spectrum of antimicrobial activity as benzylpenicillin making them useful alternatives for people with a history of penicillin and cephalosporin allergy. Erythromycin, the prototype macrolide, has been used since 1952, and clarithromycin or azithromycin are popular in treating upper respiratory infections but their administration may be accompanied by confusion, obtundation, agitation, insomnia, delirium, disorientation, psychosis and exacerbation of myasthenia gravis. The timing of these dose-dependent symptoms can range from 3 to 10 days after drug ingestion. Some may be permanent. Risk factors include psychiatric illness, renal insufficiency or excess dosage of medication [27,60,72]. Steinmam and Steinman were the first to point out visual hallucinations induced by clarithromycin taken 500 mg twice daily for acute bronchitis. This complication developed within 24 h of taking the drug. The 56-year-old patient described them as “constantly evolving landscape of sharks, priests, red lines and other technicolor” [97]. The mechanisms of CNS toxicity of macrolides are unclear. Several hypotheses include drug interactions (metabolism through isoenzyme CYP3A4), adverse effects of the lipid-soluble active metabolite of clarithromycin (14-hydroxyclarithromycin) on the CNS, alterations of cortisol and prostaglandin metabolism, as well as interactions with glutaminergic and GABA pathways [27,60,97,98,99]. Macrolides also induce ototoxicity. It has been suggested that patients may recover from transient hearing loss associated with macrolide therapy, but develop tinnitus, which may be generated in the auditory centers of the brain by deviant neuronal activity caused by macrolide use [100].

### 4.6. Aminoglycosides

Aminoglycosides are used in patients with serious gram-negative infections. They have been known to cause ototoxicity, peripheral neuropathy, encephalopathy and neuromuscular blockade [26,27,60]. Hearing loss may occur in 20–63% of patients using aminoglycosides for many days. Acute ototoxicity is related to ion channel blockade and calcium antagonism and chronic ototoxicity is based on drug access to perilymph and endolymph, and penetration of the hair cells [101]. The cause of the toxic effect on the hearing organ is the excitotoxic activation of NMDA receptors within the cochlea as a result, with subsequent oxygen radicals formulation, which is postulated to contribute to cell death [26]. The mechanism responsible for neuromuscular blockade is inhibition of quantal release of acetylcholine in the neuromuscular junction pre-synaptically and a postjunctional binding of aminoglycosides to the acetylcholine receptor complex [26,27]. Inflammation and fever increase the risk of aminoglycoside-induced hearing loss. Another cause of hearing loss may be coexisting renal insufficiency, which decreases excretion of aminoglycosides from the blood [102].

### 4.7. Oxazolidinones

This class of antibiotics is used to treat serious skin and bacterial infections, often after other antibiotics have been ineffective. They are active against a large spectrum of gram-positive bacteria, including methicillin- and vancomycin-resistant staphylococci, vancomycin-resistant enterococci, penicillin-resistant pneumococci, and anaerobes. Manifestations of oxazolidinone neurotoxicity, especially linezolid, include peripheral and optic neuropathy, serotonin syndrome, encephalopathy, and delirium. Peripheral neuropathy appears to be most commonly reported. It is more likely to occur during prolonged courses of treatment (>28 days, median 5 months). Optic neuropathies in patients treated for *Staphylococcus aureus* infections may be asymptomatic or lead to decreased visual acuity, blurred vision, central scotomas, and dyschromatopsia, The mechanism of linezolid neuropathies is unclear. It may be associated with mitochondrial injury. In addition, the drug has the ability to penetrate the central nervous and ocular system. Risk factors for developing neuropathy include pre-existing neurologic diseases, alcohol abuse, diabetes, chemotherapy and antiviral therapy. It generally improves or completely resolves after discontinuation of the medicine, although occasionally can be permanent. Linezolid is a nonselective inhibitor of monoamine oxidase. Inhibition of monoamine oxidase A increases levels of serotonin, and monoamine oxidase B elevates catecholamines. Epinephrine, norepinephrine, and dopamine are reported to be involved in serotonin syndrome, delirium or encephalopathy associated with the administration of this drug. When it is used with, e.g., selective serotonin and norepinephrine reuptake inhibitors, it can increase the risk of serotonin syndrome. Linezolid, which has dopaminergic properties, may cause serotonin syndrome if used with a monoamine oxidase inhibitor. Its administration with an anticholinergic substances increases encephalopathy risk [27,60,103,104].

### 4.8. Polymyxins

Polymyxins are peptide antibiotics of natural origin, first obtained in 1947 by fermentation in *Bacillus polymyxa* subspecies colistinus. In the early 1980s, data on the safety risks of their use related to severe episodes of renal failure, as well as incompletely understood neurotoxicity, and the availability of antibiotics with fewer potential side effects reduced their use in therapy. The incidence of neurological complications with these antibiotics ranges from 7–27%, including dizziness, generalized or muscle weakness, confusion, hallucinations, seizures, paresthesias, ataxia and, less commonly, diplopia, nystagmus and ptosis [60]. Paresthesias are more common with intravenous administration than intramuscular use. Ventilation-dependent respiratory disturbances were observed after intramuscular administration of polymyxins. They lasted from 10 to 48 h. This was probably a myasthenia-like syndrome. The polymyxin chemical structure contains a fatty acid, which may interact with the lipophilic content of neurons. Neuromuscular blockade may be related to inhibition of acetylcholine release in the synaptic cleft. Risk factors of neurotoxicity include renal dysfunction, hypoxia and concomitant use of such medication as nephrotoxic agents, sedatives, muscle relaxants, anesthetic drugs or corticosteroids [60]. Colistin neurotoxicity, especially observed in patients with renal failure or receiving high doses, includes facial paresthesias (pricking, tingling, numbness), dizziness, speech impairment, visual disturbances, confusion, and psychosis. Neuromuscular blockade manifested by myasthenia-like syndrome or as respiratory muscle paralysis producing apnea has also been observed. Colistin neurotoxicity primarily involving paresthesias, and in only sporadic cases apnea, especially in patients with intramuscular administration of the drug, with acute or chronic renal failure and receiving medications, induces respiratory muscle weakness [99,105]. Two mechanisms account for colistin neurotoxicity and neuromuscular blockade. One involves presynaptic action of the drug, preventing the release of acetylcholine into the synaptic gap. The other is biphasic, involving a short phase of competitive blockade between acetylcholine and colistin, followed by a prolonged depolarization phase, leading to loss of calcium from neurons, resulting in altered mitochondrial permeability. This results in mitochondrial dysfunction in neuronal cells and accumulation of reactive oxygen species. This in turn is the cause of oxidative stress and further nerve damage [26,106].

### 4.9. Tetracyclines

Tetracyclines are a class of broad-spectrum bacteriostatic antibiotics discovered in the 1940s, including tetracycline, minocycline, and doxycycline, which have shown to be effective against aerobic and anaerobic bacteria, as well as Gram-positive and Gram-negative bacteria (with the exceptions of *Proteus* species and *Pseudomonas aeruginosa*). They are largely prescribed in dermatology and infectious diseases, both for the anti-bacterial and anti-inflammatory actions. Neurotoxicity associated with this class of antibiotics include cranial nerve toxicity, neuromuscular blockage and intracranial hypertension [26,27,60]. During therapy with tera-cyclines, symptoms such as blurred vision, loss of balance, light-headedness, dizziness, vertigo or tinnitus were observed [60].

### 4.10. Quinolones

Quinolones are a family of antibiotics with a wide range of antimicrobial activity, which are active against both Gram-positive and Gram-negative bacteria, including mycobacteria, and anaerobes. Since their discovery in the early 1960s, they have become increasingly important in the treatment of both community and serious hospital-acquired infections. In the 1970s and 1980s, the scope of the quinolone class was greatly expanded by the groundbreaking development of fluoroquinolones, which exhibit a much broader spectrum of action and improved pharmacokinetics compared with first-generation quinolones [107]. Unfortunately, several European Medicines Agency (EMA) recommendations have recently been made to healthcare providers regarding risk factors for musculoskeletal, neurological and psychiatric adverse reactions observed among quinolone users [108].

Many members of this group of antibiotics (norfloxacin > ciprofloxacin > ofloxacin, levofloxacin) are known for their neurotoxic effects. These may manifest as headache, confusion, decline of attention, tremors, psychosis, seizures, myoclonic jerks, insomnia, encephalopathy, delirium, sleep disturbances, toxic psychosis or Tourette-like syndrome, and moreover as extrapyramidal manifestations such as gait disturbance, dysarthria and choreiform movements. Scavone et al. observed that third-generation quinolones were always associated with higher reporting probability of neurological and psychiatric adverse drug effects compared to second generation. These effects were presented 1 to 2 days after antibiotic therapy and were dose-dependent. Their etiology is likely to be multifactorial and include inhibition of GABA_A_ receptor, stimulation of NMDA receptor and ligand-gated glutamate receptors which reduce seizure threshold. It has also been suggested that oxidative stress is increased by these drugs. No less important is the relationship between their chemical structure and the symptoms observed, e.g., ciprofloxacin, norfloxacin as a quinolone with 7-piperazine and clinafloxacin, and tosufloxacin as a quinolone with 7-pyrrolidine have been observed to be highly associated with epilepsy. The epileptogenic potential of fluoroquinolones is increased by simultaneously used non-steroidal anti-inflammatory drugs (NSAIDs). Moreover, these antibiotics penetrate through the BBB and induce eosinophilic meningitis. Risk factors for neurotoxicity include older age, hypoxemia, pre-existing central nervous system diseases, electrolyte disturbance, thyrotoxicosis, renal and hepatic dysfunction. Hemodialysis may be a useful treatment for encephalopathy associated with quinolone treatment in patients with impaired renal function [27,49,72,99,108,109,110].

### 4.11. Other Antibacterial Agents (Chloramphenicol, Nitrofurantoin, Isoniazid, Ethambutol, Cycloserine)

Chloramphenicol is a broad spectrum antibiotic, which was first isolated from *Streptomyces venezuelae* in 1947. It is currently of limited use due to adverse effects and frequently observed antimicrobial resistance. It must be used only in those serious infections for which less potentially dangerous drugs are ineffective or contraindicated. Headache, mild depression, mental confusion, and delirium have been described in patients receiving this medicine. Optic and peripheral neuritis have been reported, usually following long-term therapy. If this occurs, the drug should be promptly withdrawn [26].

Nitrofurantoin, a synthetic nitrofuran derivative, has been available for the treatment of uncomplicated lower urinary tract infection since 1952. It is effective against *E. coli* and many gram-negative organisms. Nitrofurantoin treatment has been associated with neurotoxicity effects including peripheral neuropathy, dizziness, vertigo, diplopia, cerebellar dysfunction and intracranial hypertension. These are observed particularly in women and elderly patients. The etiology is attributed to axon loss [111].

Isoniazid, cyclo-serine and ethambutol-medications used for treating tuberculosis, may cause both central and peripheral neuropathy. Isoniazid administration may be accompanied by peripheral neuropathy, psychosis and seizures. The importance of isoniazid interference with GABA synthesis is emphasized in the etiology of seizures, through inhibition of pyridoxal-5 phosphate. This compound is a cofactor for the enzymatic activity of glutamic acid decarboxylase, thus reducing the concentration of GABA and enhanced seizure susceptibility. Status epilepticus was also observed after therapeutic doses of the medicine. Cyclo-serine may be the cause of neuropsychiatric adverse events including anxiety, agitation, depression, psychosis, and, rarely, seizures. The frequencies of psychiatric and central nervous system adverse events are 5.7 and 1.1%, respectively. They may be associated with elevated plasma concentrations of the drug. Cyclo-serine crosses the blood–brain barrier and decreases GABA production. It binds to N-methyl-d-aspartate receptors, which in part explains the commonly associated neurotoxicity. At the recommended dosage for cyclo-serine (250 to 500 mg once daily), the neurotoxicity can range from mild to severe and has resulted in psychosis and treatment discontinuation in some cases. Concurrent use of alcohol increases the risk of developing psychosis and seizures. Another complication of ethambutol therapy can be optic nerve neuropathy. This is dose-dependent, with the lowest risk at total daily doses < 15 mg/kg. Its risk factors include older age, hypertension, renal insufficiency, and duration of treatment. Symptoms are manifested by gradual onset of reduced visual acuity, dyschromatopsia and central or mid-central visual field losses observed several months after the drug was started. They are probably related to mitochondrially induced papillary bundle dysfunction [27,99,112].

To sum up, the mechanisms contributing to the neurotoxic adverse effects of antibiotics are multiple and specific to a given class of those drugs. They are summarized in Table 3 below.

## 5. Methods of Reducing the Frequency and Severity of Antibiotic-Induced Neurologic and Psychiatric Entities

Modern optimal antibiotic therapy requires extensive knowledge of the mechanisms of drug action, their pharmacokinetic properties, adverse effects, identification of their risk factors, especially underestimated neurotoxicity, toxicity thresholds limiting dosing, infection site and antibiotic penetration, and careful monitoring of the consequences of their action. It is necessary to control the clinical condition of patients, to examine the efficiency of organs responsible for elimination of drugs from the body. Early recognition of renal failure may reduce the frequency or severity of neurologic and psychiatric symptoms associated with antibiotic administration. EEG may be helpful in differentiating between drug complications in the form of non-convulsive status epilepticus (NCSE) and encephalopathy. Sometimes temporary use of anti-convulsant medication may be needed. Myasthenic syndrome accompanying treatment with polymyxins may require ventilatory support depending on the degree of respiratory impairment. Hemodialysis or hemofiltration may be needed in patients with impaired renal function if antibiotic-induced neurotoxicity is observed [26,113].

In recent years, much attention has been given to increasing the optimization of antibiotic therapy based on pharmacokinetic and pharmacodynamic (PK/PD) modelling [114,115,116,117]. To evaluate the efficacy and safety of antimicrobial therapy, three basic ratios were developed: Cmax/MIC (minimal inhibitory concentration), T > MIC, AUC_24_/MIC. Concentration-dependent antibiotics include aminoglycosides and metronidazole. Their efficacy best correlates with peak concentration (Cmax) to MIC. Clinical PK/PD target for amikacin/gentamicin efficacy is Cmax/MIC ≥ 8–10, clinical PK/PD threshold for amikacin toxicity is Cmin > 5 mg/L, for gentamicin > 1 mg/L. The group of antibiotics whose effectiveness is determined by the time the concentration remains above the MIC of the bacterial pathogen include penicillin, cephalosporins, carbapenems, monobactams, macrolides (erythromycin, clarithromycin), linezolid. Clinical PK/PD target for carbapenems/penicillin efficacy is 50–100% fT>MIC, for cephalosporins 45–100%, clinical PK/PD threshold for meropenem nephro- or neurotoxicity is Cmin > 44.5–64 mg/L, for neurotoxicity of cefepime Cmin ≥ 20–22 mg/L, for piperacillin Cmin > 64–361 mg/L. Concentration-dependent antibiotics with a time-dependent component for which the best predictor of efficacy is the area under the concentration-time curve during a 24 h time period (AUC_24_) to the MIC ratio include: glycopeptides, oxazolidinones, fluoroquinolones, polymixins, daptomycin, azithromycin and tigecycline. Clinical PK/PD target for vancomycin efficacy is AUC_0–24_/MIC ≥ 400, threshold for its toxicity AUC_0–24_ > 700 mg·h/L, Cmin > 20 mg/L [118]. This individualized approach has allowed two directions for optimizing antibiotic therapy, especially in intensive care patients: dose adjustment based on therapeutic drug monitoring (TDM) or modification of drug dosing by using higher initial and maintenance doses or by using prolonged or continuous infusions [119,120,121].

TDM, the measurement of drug concentrations in biological fluid (typically plasma) is particularly important with respect to drugs with a narrow therapeutic index, with a defined relationship between their concentration and pharmacological effect, significant intra-and/or inter-individual pharmacokinetic variability, established target concentration range, which are the cause of numerous drug complications and interactions with other drugs, long duration of therapy, absence of pharmacodynamic markers of therapeutic response and/or toxicity, and availability of cost-effective drug assay (precise, accurate, highly selective bioanalytical assay methods for drug measurement). It is widely used for aminoglycosides, and vancomycin, for beta-lactam antibiotics, particularly for piperacillin and meropenem, is becoming increasingly common [113,122,123]. It is important to remember that the drug concentration is only complementary but not a substitute for clinical judgement, and we treat the individual patient, not the laboratory value.

Imami et al. retrospectively reviewed a series of cases of people treated with potentially neurotoxic antibiotics hospitalized at St Vincent Hospital in Sydney between 2013 and 2015. Adverse events of neurotoxicity, nephrotoxicity, hepatotoxicity and *Clostridium difficile* infections were assessed. Based on the measurements of drug concentrations (piperacillin, meropenem, fluo-cloxacillin), their direct relationship with the complication was demonstrated. The breakpoint for which the risk of neurotoxicity is 50% for piperacillin was found to be Cmin > 361.4 mg/L, for meropenem > 64.2 mg/L, and for flucloxacillin > 125.1 mg/L. Therefore, measuring the concentrations of these antibiotics, especially in patients with an increased risk of neurological complications, is a method of optimizing their use [124].

Oda et al. reported a case of using Bayesian estimation calculations in conjunction with the measurement of cefepime concentration to reduce the dose in people with pneumonia to prevent neurological complications. After receiving a dose of 1.0 g every 8 h, the patient developed aphasia on the fifth day. Measurement of the drug concentration in the serum showed 71.3 mg/l, which was 2–3 times higher than the recommended value (22–35 mg/L). Bayesian pharmacokinetic calculations indicated the need to reduce the dose to 0.5 g every 12 h. After 3 days, the neurological symptoms improved and the treatment was continued successfully [125].

Another case was described by Smith et al. and concerned an 82-year-old patient admitted to the intensive care unit with a diagnosis of severe community-acquired pneumonia, septic shock and multiple organ failure. After administration of cefepime, the patient developed convulsions. Blood and cerebrospinal fluid drug concentrations were measured and increased values were found. After dose adjustments and a decrease in cefepime levels, the seizures subsided [126].

In 2020, a summary of an expert discussion panel on the use of TDM in relation to antibiotics, antifungal and antiviral drugs in intensive care units was published. It was emphasized that from a clinical practice point of view dosing drugs under TDM control is beneficial for the aminoglycosides, voriconazole and ribavirin. Therapeutic ranges have been defined for some of the antibiotics. Routine use of TDM has been recommended for therapy with aminoglycosides, beta-lactam antibiotics, linezolid, teicoplain, vancomycin and voriconazole in critically ill patients. The authors pointed out that, although drug concentration monitored therapy was first used in the 1940s, it still requires the development of globally uniform standards of care, especially with regard to the treatment of patients with comorbidities and multi-organ disorders [118].

According to a systematic review by Barreto et al., clinical observations have shown that, in critically ill patients, beta-lactam antibiotic levels must be monitored, and the recommended minimum concentrations should be greater than the MIC for at least half the time between doses. The free drug fraction is recommended to be measured during the first 48 h of therapy, and should be above the MIC breakpoint of the most likely pathogen before blood culture results are available. This concentration should be maintained for the entire period between doses, and after this time the minimum concentration should reach a value of 1–2x of the observed MIC of the pathogen obtained in microbiological cultures. Neurotoxicity has also been shown to be the most dose-dependent adverse event, although direct evidence is not yet available to indicate the concentration above which this complication is likely to occur [127].

A retrospective cohort study published in 2017, which included 53 patients admitted to the intensive care unit with no neurological abnormalities prior to commencing continuous infusion of piperacillin at the standard dose and subjected to serum piperacillin determination, showed that 23 patients developed a neurological disorder, in which piperacillin causation was consistent chronologically and semiologically. The minimum concentration value of 157.2 mg/L, regardless of other variables, was the factor of the occurrence of neurotoxicity with 96.7% specificity and 52.2% sensitivity. This is a phenomenon that may be a limitation in antibiotic therapy if the patient has pathogens less sensitive to this antibiotic [128].

Optimization of antibiotic therapy also requires the use of guidelines adapted to local needs and adherence to these by medical staff. Unfortunately, a multi-center study has shown that 37.8% of antibiotic use in European hospitals does not comply with this restriction. Antimicrobial stewardship programs are a promising strategy. One of the methods is a pharmacokinetic dosing nomogram. This describes the influence of a covariate (e.g., weight) on a drug exposure target (e.g., concentration). It can be combined with TDM. Clinical pharmacological advice, which is delivered by the clinical pharmacologist who interprets the therapeutic drug monitoring results of antimicrobials in relation to the site of infection, the pathophysiological characteristics of the patient, and potential drug–drug interactions are very important in personalized treatment [129,130]. The interprofessional team should include a clinical pharmacist, who can play an important role by monitoring antimicrobial prescriptions and providing advice or educating medical and nursing staff, because approximately 50% of hospitalized patients receive at least one antibiotic, and 20-30% cases of antibiotic therapy are unnecessary. Clinical pharmacist intervention has been shown to be effective in enhancing appropriate use of antibiotics and reducing their toxicity, which may improve patient care. Moreover these had a positive impact not only on the clinical, but on financial outcomes [131,132,133,134,135].

## 6. Neuroprotective Action of Antibiotics

In recent years, old, well-known drugs have been increasingly used in new indications. Such a strategy is referred to as “repositioning drugs”, “redirecting drugs” or “finding new uses for old drugs”. It is an efficient and cost-effective pathway to new drug development. Antibiotics are also being studied for their anti-amyloidogenic and anti-inflammatory properties. Results of ongoing observations suggest the possible use of antibiotics in Alzheimer’s disease, Parkinson’s disease or multiple sclerosis. Tetracyclines, and especially doxycycline, are promising in this area. Interest in their use in Alzheimer’s disease dates back to the early 2000s when it was discovered that tetracyclines could inhibit the aggregation of the β-amyloid peptide (Aβ). Moreover they have anti-oxidative and anti-apoptotic activities [136,137,138]. Many studies have confirmed the neurotrophic, anti-inflammatory, antioxidant and anti-apoptotic effects of minocycline, a long-acting, semi-synthetic tetra-cycline. This antibiotic is characterized by high lipophilicity and can easily penetrate the blood–brain barrier, has long half-life time and excellent tissue penetration. It alters the reactivity of microglia cells, counteracts inflammatory processes, and reduces neurodegenerative processes within the central nervous system. Its effectiveness has been proven in experimental models for the treatment of Alzheimer’s disease, Parkinson’s disease, Huntington’s disease, multiple sclerosis, neuropathic pain, stroke, hypoxic-ischemic encephalopathy and hypomyelination. It has been shown to reduce white matter and hippocampal lesions and improve cerebral blood flow. The drug reduces the expression of pro-inflammatory markers responsible for increasing the activity of chemokine CCL2, IL-1β, IL-6, TNF-α and iNOS. Minocycline has antioxidant and antiapoptotic properties, manifested by caspase inhibition. In turn, ceftriaxone was found to increase the expression of astrocytic glutamate transporter 1 (GLT-1), decreasing excitotoxicity and neuroinflammation by detoxifying the brain from glutamate. It should be emphasized that persistently elevated amounts of this compound in the synaptic space may contribute to neurodegenerative diseases and ischemic stroke [139]. It affects the markers of oxidative status and neuroinflammation [138]. Rifampicin is also a broad-spectrum antibiotic whose protective effect on the brain has been demonstrated in many experimental studies. Its mechanism of action includes inhibitory effect on free oxygen radicals, tau and Aβ protein accumulation, microglial activation, apoptotic cascades [140]. Use of antibiotics in neuroprotection is promising, creates new potential treatment options for neurodegenerative diseases, but requires many more studies using not only laboratory models but also human subjects.

## 7. Conclusions

The study of neuroprotective drugs that lead to rescue, recovery or regeneration of the nervous system, its cells, structure and function, has been ongoing for many years. These are based on three main strategies, i.e., the synthesis of new drugs, the use of natural products with as yet unidentified properties, and attempts to develop therapies based on existing drugs, so-called “drug repositioning” or “drug reprofiling”. The latter area seems worthy of attention, as the pharmacokinetic and pharmacodynamic profile of such drugs is already known, and the effort put into such a strategy requires incomparably less time and cost than developing new drugs. Unfortunately, this is not an easy task, as there are many neurochemical modulators of nervous system damage. Clinical trials often fail to demonstrate their efficacy, and the doses used prove toxic [141]. Patients with nervous system dysfunctions are also a very heterogeneous group in terms of both their etiology and their age, etc., and they are additionally burdened with various risk factors. Experimental models also differ significantly from clinical conditions. The development of such drugs requires a better understanding of the etiology and pathogenesis of nervous system diseases. It is believed that neuroinflammatory mechanisms may account for many of the processes responsible for the neuronal degeneration observed in Alzheimer’s disease, Parkinson’s disease, stroke, and other neurodegenerative diseases. They undoubtedly represent a significant health problem and challenge for 21st century medicine. Antibiotics are also being investigated in this aspect and promising observational results provide new potential avenues for their use as neuroprotective rather than just anti-infective drugs. Rifaximin is currently in phase II clinical trials based on the association between changes in the gut microbiota and neuropsychiatric diseases. It is hypothesized that it may improve memory and daily functioning in people with Alzheimer’s disease by reducing blood levels of ammonia and/or levels of pro-inflammatory cytokines secreted by gut bacteria [142]. On the other hand, it is very important to pay attention to the possibility of neurotoxicity during antibiotic therapy. Multidirectional monitoring of patients at high risk of neurotoxicity is necessary to prevent or reduce its severity. As described above, its causes are not fully understood. It is also necessary to conduct multidirectional research dedicated to the elucidation of mechanisms responsible for nervous system dysfunction under the influence of antimicrobial drugs. To achieve an effective antimicrobial effect, and at the same time not to induce drug-related complications, the choice of antibiotic and therapy depend on clinical diagnosis, pathogens isolated from patient or those most frequently causing a specific infection in a population and their sensitivity to antibiotics, concomitant diseases present in the patient (taking into account past diseases, chronic diseases, impaired renal or hepatic function, age, allergies, etc.), and properties of the antibiotic itself (pharmacodynamics, pharmacokinetics, possible side effects, toxicity).

## Figures and Tables

**Table 1 molecules-26-07456-t001:** Examples of drug-induced neurological disorders (DIND).

Disorder/Syndrome	Symptoms	Drugs
Cerebrovascular disorders	Stroke due to deep venous thrombosis or pulmonary embolism Cerebellar syndrome	estrogens/progestins (oral contraceptives) antiepileptic drugs (phenytoin, carbamazepine), lithium, selected antibiotics
Cognitive impairment and delirium	Dementia Fluctuations in cognition, mood, attention and arousal	1-st generation antihistamines, antiparkinsonian agents, skeletal muscle relaxants, tricyclic antidepressants, antipsychotics, benzodiazepines
Neuroleptic malignant syndrome	Muscular rigidity, tremor, possible muscle tissue breakdown, autonomic instability, high fever, changes in cognition	antipsychotics (neuroleptics)
Nerve and muscle disorders	Muscular weakness, loss of coordination, possible paralysis	benzodiazepines selected antibiotics
Movement disorders	Akathisia, dystonia, pseudo-parkinsonism	dopamine receptor blockers: 1-st generation neuroleptics and antiemetics (metoclopramide), anticholinergic agents (benztropine, diphenhydramine), benzodiazepines
Epilepsy	Seizures or impairment of consciousness and/or movements	benzodiazepines (when suddenly withdrawn), diuretics (due to electrolyte imbalance), antiarrhythmics, bupropion, antipsychotics (chlorpromazine, clozapine), lithium, opiate analgesics (fentanyl, meperidine, tramadol), selected antibiotics
Serotonin syndrome	Cognitive and behavioral changes, autonomic instability, high blood pressure, sweating, agitation, tremor, fever, nausea and vomiting	serotonin reuptake inhibitors serotonin-norepinephrine reuptake inhibitors, tricyclic antidepressants, opiate analgesics (meperidine, dextromethorphan), anti-migraine drugs-triptans, selected antibiotics
Sleep disorders	Insomnia or excessive daytime sleepiness with decreased ability to concentrate, think and reason	stimulants: adrenergic agents, antidepressants, corticosteroids, antiparkinsonian agents, sleep-inducing agents (when overused or suddenly discontinued)
Disorders of the sense organs	Hearing and vision impairment	selected antibiotics

**Table 2 molecules-26-07456-t002:** Possible and most common adverse drug reactions in the form of neurotoxicity of different classes of antibiotics.

Class of Antibiotic	Neurotoxicity
penicillin	confusion, disorientation, tardive seizure, encephalopathy, tremors
cephalosporins	lethargy, tardive seizures, myoclonus, encephalopathy, chorea, athetosis,
carbapenems	headache, seizures, encephalopathy, myoclonus, peripheral neuropathy
glycopeptides	ototoxicity
macrolides	ototoxicity, seizures, confusion, agitation, insomnia, delirium, exacerbation of myasthenia gravis
aminoglycosides	ototoxicity-class effect, peripheral neuropathy, neuromuscular blockade class-effect, autonomic dysfunction
oxazolidinones	encephalopathy, peripheral neuropathy, optic neuropathy
polymyxins	Encephalopathy, paresthesias, ataxia, diplopia, potosís and nystagmus, vertigo, confusion, ataxia, seizures
tetracyclines	cranial nerve toxicity, neuromuscular blockade, intracranial hypertension
lincosamides	movement disturbances
chloramphenicol	optic neuropathy
sulfonamides /trimethoprim	tremor, transient psychosis, encephalopathy, aseptic meningitis
quinolones	headache, seizures, confusion, insomnia, encephalopathy, myoclonus, orofacial dyskinesias, ataxia, chorea, extra-pyramidal disturbances
metronidazole	headache, dizziness, confusion, encephalopathy, optic neuropathy, peripheral neuropathy
nitrofurantoin	intracranial hypertension, peripheral neuropathy
isoniazid, ethambutol, cyclo-serine	peripheral neuropathy, seizures, optic neuropathy

**Table 3 molecules-26-07456-t003:** Summary of the mechanisms of neurotoxicity of particular classes of antibiotics.

Class of Antibiotic	Mechanisms of Neurotoxicity
penicillin	GABA complex receptor inhibition via competitive or non-competitive affecting the GABA_A_ subunits; an increase of the N-methyl-D-aspartate (NMDA) and alpha-amino-3-hydroxy-5-methylisoxazolepropionate receptors stimulation resulting in the overactivity of glutamatergic system
cephalosporins	
glycopeptides	direct damage of the auditory branch of the eighth cranial nerve; an increase of the oxidative stress leading to loss of sensory cochlear cells
macrolides	drug interactions (metabolism through isoenzyme CYP3A4); direct neurotoxic effect produced by the lipid-soluble active metabolites; alterations of cortisol and prostaglandin metabolism; interactions with glutaminergic and GABA pathways
aminoglycosides	Ototoxicity-determined by the overactivation of NMDA receptors within the cochlea with subsequent oxygen radicals formulation; neuromuscular blockade-due to the presynaptic inhibition of quantal release of acetylcholine in the neuromuscular junction and a postjunctional blockade of the acetylcholine receptor complex
oxazolidinones	mitochondrial injury; nonselective inhibition of monoamine oxidase leading to increased serotonin and catecholamines levels
polymyxins	neuromuscular blockade-due to the presynaptic decrease of acetylcholine release into the synaptic gap; induction of prolonged depolarization following the transient postsynaptic blockade, with loss of calcium from neurons and altered mitochondrial permeability; accumulation of reactive oxygen species
quinolones	inhibition of GABA_A_ receptor; stimulation of NMDA receptor and ligand-gated glutamate receptors; an increase of the oxidative stress
sulfonamides /trimethoprim	deficiency in the tetrahydrobiopterin synthesis resulting in disturbances in synthesis of central neurotransmitters
metronidazole	an increase of oxidative stress; oxidation of catecholamines and other neurotransmitters; inhibition of GABA-ergic neurotransmission
other anti-infective agents: nitrofurantoin, isoniazid, ethambutol	loss of axons; decrease of GABA synthesis, NMDA receptors activation

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
