# Peer review of "Antibiotics and the Nervous System—Which Face of Antibiotic Therapy Is Real, Dr. Jekyll (Neurotoxicity) or Mr. Hyde (Neuroprotection)?"

_molecules, 2021, doi:10.3390/molecules26247456_

Round 1

Reviewer 1 Report

The review titled “Which Face of Antibiotic Therapy is Real, Dr Jekyll or Mr Hyde?” shows how antibiotics can have a dual biological action on our bodies. As is well known, the widespread abuse of antibiotics can cause bacterial resistance and alter the gut microbiome, causing adverse neurological/psychiatric reactions. However, antibiotics have also been studied for their neuroinflammatory and neuroprotective properties in the treatment of neurodegenerative processes such as Alzheimer’s disease.

This report is interesting as it provides new insights into the promising elements of multi-targeted therapy. The introduction is appropriate, and the aims of the review are well displayed.

The manuscript is suitable for publication in its current form, with the following minor modifications:

• The title is smart but does not clearly and accurately reflect the content of the manuscript; in fact, most of the paragraphs of this review are devoted to the harmful aspects of antibiotic treatment (Dr Jekyll), while only a short final paragraph focuses on its positive aspects (Mr Hyde).

• A summary table is also needed for paragraph n. 4. “A short description of the detailed neurotoxicity of particular classes of antibiotics.”

• The conclusions are too brief; please expand them by providing your own conclusive interpretation of the study.

Otherwise, this is an adequate, coherent review.

Author Response

Author’ reply to the comments of the Reviewers

We would like to thank you for your kind evaluation of our manuscript and your valuable comments enabling the improvement the quality of our text. Please find below a detailed list of the changes made.

Reviewer 1

Dear Reviewer, thank you for your valuable comments. In the current form of our manuscript, we tried to introduce amendments which were to meet your suggestions.

(… ) The manuscript is suitable for publication in its current form, with the following minor modifications:

  • The title is smart but does not clearly and accurately reflect the content of the manuscript; in fact, most of the paragraphs of this review are devoted to the harmful aspects of antibiotic treatment (Dr Jekyll), while only a short final paragraph focuses on its positive aspects (Mr Hyde).

In line with your suggestion, we changed the title of our manuscript. In the current version, the title of our paper is: “Antibiotics and the nervous system - Which face of antibiotic therapy is real, Dr Jekyll (neurotoxicity) or Mr Hyde (neuroprotection)?

  • A summary table is also needed for paragraph n. 4. “A short description of the detailed neurotoxicity of particular classes of antibiotics.”

Dear Reviewer, a brief summary of the neurotoxic effects produced by antibiotics is given in Table 2. It lists the neurological adverse effects of the various class of antibiotics discussed in our manuscript. In our intention, Table 2 was to precede a detailed description of the neurotoxicity of the following groups of antibiotics given in Chapter 4. However, according to your recommendation, new Table 3 was introduced which summarizes the mechanisms of neurotoxicity, discussed in Chapter 4

  • The conclusions are too brief; please expand them by providing your own conclusive interpretation of the study.

Thank you for your suggestion, we changed this part of manuscript

Otherwise, this is an adequate, coherent review.

Thank you for your kind opinion.

Reviewer 2 Report

Thank you for your manuscript. I have a few suggestions

Line 49 change "into" to "in"

Line 75-77. The sentence starting with "An adverse drug reaction is regarded to be an unwanted, harmful, or unpleasant effect...."  Please incorporate "expected" reaction into the sentence. Selection of antimicrobial therapy takes into account the likely and potential ADR associated with the medication.

Line 97. Change Fluorochinolone to fluoroquinolone.

Line 99. Delete "(mostly by erthromycin)". It isn't required. 

Line 115. Add polymixins  to the list of antimicrobials that can impact the nervous system.

Table 1. Under Disorder/Syndrome - Serotonin Syndrome. Add selected antibiotics under drugs. Linezolid listed on line 497 references the potential for serotonin syndrome. 

Line 35 and 138. add the superscript 2+ for Ca and + for Na.

Line 165-167. Revise the sentence that incorporates increased/prolonged duration of high doses of antibiotic increase the risk for neurotoxicity.

Line 173.  Change Fluorochinolone to fluoroquinolone.

Line 175. Reference 18. Please clarify this reference. There does not appear to be discussion of "mumble speech" in the paper. 

Line 318. Change trimetoprim to trimethoprim

Line 321. Change, "currently, sulfonamides have lost their importance, but..." to " Currently, sulfonamides have diminished importance due to resistance, but are still used in the treatment of certain infections...."

Line 358. Add the word "commonly".  SMX/TMP neurotoxicity is less commonly reported in children.... 

Line 399. Remove the "e" from cefazolin.

Line 407. use a subscript for A in GABA A

Line 423. Delete "up to date,"

Line 426. Reference 71 states that the rates of neurological ADRs were from 0-0.6% in clinical trials, which contradicts the sentence that states "their frequency was not specified."

Line 428. Delete "gamma-aminobutyric acid." Use GABA only as it was already referenced as a neurotransmitter on line 250.  If you want to keep it then I suggest you move the definition of the term either to line 250 or 406 where GABA is used previously.

Line 452-454. change "dalbavancin and oritavancin are second-generation...." to Currently available data suggests that the second generation lipoglycopeptides, dalbavancin and oritavancin, have no effect on hearing loss or dysfunction."  The rationale for the change is because the available data is too small in number and there isn't enough evidence to suggest that is is 100% risk free.

Line 454.  Reference 96 refers to delafloxacin and not oritavancin. Another reference is needed in place of reference 96.

Line 481. Change "Chronic one" to something more appropriate. The original text from the citation used "Chronic mechanism are..."  I am not sure if "one" was a typo or another word. 

Line 487.  Fix the reference [26,27l] to [26,27]

Line 492-495. Remove the underlining. It is not necessary. 

Line 553. Reference 60 has a lower cased L after it. Please remove.

Line 575. Change "have shown from" to "manifests in" or typically presents in"

Line 589. There is extra close bracket (]).

Line 618. Change "At the currently recommended dosing..." to At the recommended dose for cycloserine (250 to 500 mg once daily), the ...

Line 658. Reference 64 is not correct for that citation. Reference 64 is for "A rare case of cotrimoxazole-induced eosinophilic aseptic meningitis in an HIV-infected patient. Scand J Infect Dis. 1998". Please insert a more appropriate reference.

Line 688. change "meropfor nem>64.2" to "for meropenem>64.2"

Line 770. Hunigton's disease. to Huntington's disease?

Lastly, Line 789-793, the conclusion. I think the conclusion needs work. It doesn't reflect the 6 major sections in the paper.

Author Response

Author’ reply to the comments of the Reviewers

We would like to thank you for your kind evaluation of our manuscript and your valuable comments enabling the improvement the quality of our text. Please find below a detailed list of the changes made.

Reviewer 2

Thank you for your manuscript. I have a few suggestions.

Dear Reviewer, thank you for your valuable comments. Below we present the introduced changes and we hope they meet your expetations.

Line 49 change "into" to "in"

The suggested amendment has been included in the current version of the manuscript

Line 75-77. The sentence starting with "An adverse drug reaction is regarded to be an unwanted, harmful, or unpleasant effect...."  Please incorporate "expected" reaction into the sentence. Selection of antimicrobial therapy takes into account the likely and potential ADR associated with the medication.

The suggested amendment has been included in the current version of the manuscript

Line 97. Change Fluorochinolone to fluoroquinolone

We apologize for the linguistic mistake, it has been corrected

Line 99. Delete "(mostly by erthromycin)". It isn't required.

The suggested amendment has been included in the current version of the manuscript

Line 115. Add polymixins  to the list of antimicrobials that can impact the nervous system.

The suggested amendment has been included in the current version of the manuscript

Table 1. Under Disorder/Syndrome - Serotonin Syndrome. Add selected antibiotics under drugs. Linezolid listed on line 497 references the potential for serotonin syndrome. 

The suggested amendment has been included in the current version of the manuscript

Line 35 and 138. add the superscript 2+ for Ca and + for Na.

The suggested amendment has been included in the current version of the manuscript

Line 165-167. Revise the sentence that incorporates increased/prolonged duration of high doses of antibiotic increase the risk for neurotoxicity.

In the current version, the previous sound of the sentence “The risk of neurotoxicity increases with high doses of administered antibiotics and patients with abnormal renal and/or kidney functions may experience neurotoxicity at lower doses” has been reworded: “The antibiotic-related neurotoxicity depends on the dosing schedule and the functional status of the liver and kidneys.”

Line 173.  Change Fluorochinolone to fluoroquinolone.

We apologize for the linguistic mistake, it has been corrected

Line 175. Reference 18. Please clarify this reference. There does not appear to be discussion of "mumble speech" in the paper. 

Dear Reviewer, we apologize for the imprecise description of the speech disorder as "mumble speech" in the previous version of the manuscript. The citied item No. 18, which the interested reader can find a reference to the problem in, has also been changed and the indicated sentence was reworded. In the current version it sounds:

(…) The other, unusual effects observed in patients treated with fluoroquinolones (ofloxacin, sparfloxacin) included orofacial dyskinesia and a Tourette-like syndrome [18].

[18] Ruiz, M.E.; Wortmann, G.W. Unusual effects of common antibiotics. Clevel. Clin. J. Med. 2019, 86, 277-281

Line 318. Change trimetoprim to trimethoprim

We apologize for the linguistic mistake, it has been corrected

Line 321. Change, "currently, sulfonamides have lost their importance, but..." to " Currently, sulfonamides have diminished importance due to resistance, but are still used in the treatment of certain infections...."

Thank you for your reformulation of the sentence, it was introduced into the text

Line 358. Add the word "commonly".  SMX/TMP neurotoxicity is less commonly reported in children.... 

Thank you for your amendment, it has been included in the text

Line 399. Remove the "e" from cefazolin.

We apologize for the linguistic mistake, it has been corrected

Line 407. use a subscript for A in GABA A

The suggested amendment has been included in the current version of the manuscript.

Line 423. Delete "up to date,"

The suggested amendment has been included in the current version of the manuscript.

Line 426. Reference 71 states that the rates of neurological ADRs were from 0-0.6% in clinical trials, which contradicts the sentence that states "their frequency was not specified."

The phrase “their frequency was not specified has been deleted”

Line 428. Delete "gamma-aminobutyric acid." Use GABA only as it was already referenced as a neurotransmitter on line 250.  If you want to keep it then I suggest you move the definition of the term either to line 250 or 406 where GABA is used previously.

The suggested amendment has been made and the abbreviation “GABA” was used in the indicated fragment.

Line 452-454. change "dalbavancin and oritavancin are second-generation...." to Currently available data suggests that the second generation lipoglycopeptides, dalbavancin and oritavancin, have no effect on hearing loss or dysfunction."  The rationale for the change is because the available data is too small in number and there isn't enough evidence to suggest that is 100% risk free.

Thank you for your reformulation of the sentence, it was introduced into the text

Line 454.  Reference 96 refers to delafloxacin and not oritavancin. Another reference is needed in place of reference 96.

Reference 96 refers to oritavancin: Townsend, M.L.; Wilson, D.; Pound, M.; Drew, R. Emerging treatment options for complicated skin and skin structure infections: oritavancin. Clin Med Insight: Therapeutics 2010, 2, e.10.4137

Line 481. Change "Chronic one" to something more appropriate. The original text from the citation used "Chronic mechanism are..."  I am not sure if "one" was a typo or another word. 

Thank you for your amendment. Now, in the text is: Acute ototoxicity is related to ion channel blockade and calcium antagonism and chronic ototoxicity is based on drug access to perilymph and endolymph, penetration in the hair cells.

Line 487.  Fix the reference [26,27l] to [26,27]

We apologize for the mistake, it has been corrected

Line 492-495. Remove the underlining. It is not necessary. 

We apologize for the mistake, it has been corrected

Line 553. Reference 60 has a lower cased L after it. Please remove.

We apologize for the mistake, it has been corrected

Line 575. Change "have shown from" to "manifests in" or typically presents in"

Thank you for your amendment, it has been included in the text

Line 589. There is extra close bracket (]).

We apologize for the mistake, it has been corrected

Line 618. Change "At the currently recommended dosing..." to At the recommended dose for cycloserine (250 to 500 mg once daily), the ...

Thank you for your amendment, it has been included in the text

Line 658. Reference 64 is not correct for that citation. Reference 64 is for "A rare case of cotrimoxazole-induced eosinophilic aseptic meningitis in an HIV-infected patient. Scand J Infect Dis. 1998". Please insert a more appropriate reference.

It isn't a reference but concentrations, the notation was changed

Line 688. change "meropfor nem>64.2" to "for meropenem>64.2"

We apologize for the mistake, it has been corrected

Line 770. Hunigton's disease. to Huntington's disease?

We apologize for linguistic mistake, it has been corrected

Lastly, Line 789-793, the conclusion. I think the conclusion needs work. It doesn't reflect the 6 major sections in the paper.

Thank you for your suggestion, we changed this part of manuscript.

Reviewer 3 Report

egarding the manuscript, the author try to show the "forgotten" antibiotic side effect: neurotoxicity. The manuscript is well written, has suitable scientific sound, and it is easy to read.  In my opinion, the manuscript must address just one suggestion:

The conclusion is too short and not taken into account the last part of this: Neuroprotective action of antibiotic. Besides, the real face of the antibiotic it is not revealed in the conclusion.

After this consideration, the manuscript must be published.

My sincerely,

Author Response

Author’ reply to the comments of the Reviewers

We would like to thank you for your kind evaluation of our manuscript and your valuable comments enabling the improvement the quality of our text. Please find below a detailed list of the changes made.

Reviewer 3

Regarding the manuscript, the author try to show the "forgotten" antibiotic side effect: neurotoxicity. The manuscript is well written, has suitable scientific sound, and it is easy to read.  In my opinion, the manuscript must address just one suggestion:

The conclusion is too short and not taken into account the last part of this: Neuroprotective action of antibiotic. Besides, the real face of the antibiotic it is not revealed in the conclusion.

Thank you for your suggestion, we changed this part of manuscript.

After this consideration, the manuscript must be published.

Thank you for your kind evaluation of our manuscript.